# Reinforcement Learning-Based Autonomous Driving at Intersections in CARLA Simulator

**DOI:** 10.3390/s22218373

**Published:** 2022-11-01

**Authors:** Rodrigo Gutiérrez-Moreno, Rafael Barea, Elena López-Guillén, Javier Araluce, Luis M. Bergasa

**Affiliations:** Department of Electronics, University of Alcalá, 28805 Alcalá de Henares, Spain

**Keywords:** reinforcement learning, decision making, autonomous driving

## Abstract

Intersections are considered one of the most complex scenarios in a self-driving framework due to the uncertainty in the behaviors of surrounding vehicles and the different types of scenarios that can be found. To deal with this problem, we provide a Deep Reinforcement Learning approach for intersection handling, which is combined with Curriculum Learning to improve the training process. The state space is defined by two vectors, containing adversaries and ego vehicle information. We define a features extractor module and an actor–critic approach combined with Curriculum Learning techniques, adding complexity to the environment by increasing the number of vehicles. In order to address a complete autonomous driving system, a hybrid architecture is proposed. The operative level generates the driving commands, the strategy level defines the trajectory and the tactical level executes the high-level decisions. This high-level decision system is the main goal of this research. To address realistic experiments, we set up three scenarios: intersections with traffic lights, intersections with traffic signs and uncontrolled intersections. The results of this paper show that a Proximal Policy Optimization algorithm can infer ego vehicle-desired behavior for different intersection scenarios based only on the behavior of adversarial vehicles.

## 1. Introduction

Intersections in urban environments are one of the most complex and challenging scenarios in autonomous driving. These scenarios are accident-prone; numerous traffic crash statistics show that 60% of severe traffic injuries in Europe are related to intersections [1]. In the US, the National Highway Traffic Safety Administration (NHTSA) publishes an annual report related to traffic safety [2]. These data show that 29% of all car crashes and 18% of pedestrian fatalities occur at intersections. For these reasons, an autonomous vehicle needs to make safe and efficient decisions and develop reliable driving strategies [3].

One of the main reasons for this high accident rate is the large amount of information that the driver perceives from the environment. For example, in an intersection, the driver has to be aware of traffic signals such as stop signs or traffic lights and at the same time has to monitor other vehicles and estimate their velocities and intentions. This requires great attention to driving when making a decision. Therefore, developing an agent that allows safe and reliable decisions is a hard task to implement manually [4].

In the last few years, several strategies have already been applied to find a solution for the intersections challenge. These approaches are mainly focused on making decisions considering motion prediction and collaborations by applying vehicle-to-vehicle (V2V) and vehicle-to-infrastructure (V2I) technologies [5,6]. Currently, the most widely used agent is based on time-to-collision (TTC). This parameter has been used in multiple works as a safety indicator, both in research and the automotive industry [7,8]. The main problem is that some tuning parameters have to be adjusted, and this task can be laborious because of its dependence on environmental situations. As a result, this agent requires several rules to deal with different scenarios in an urban environment, and therefore, it is not reliable enough for autonomous driving. On the other hand, several machine learning based approaches have been implemented to tackle the intersection problem. These approaches include imitation learning, where the policy is learned from a human driver, and reinforcement learning, where an agent learns in an interactive environment by trial and error using feedback from its actions and experiences [9,10].

In this paper, we provide a Deep Reinforcement Learning (DRL) approach for intersection handling that combines DRL with Curriculum Learning (CL) [11]. Our goal is to train an algorithm that can autonomously learn to solve different types of intersections and accelerate the training by incorporating reinforced CL into the process. To do this, we propose a two-stage training method. A simple state representation and a light simulator such as SUMO [12] allow a fast first training stage. The obtained model is used as a prior in a second training stage in a realistic CARLA simulator where vehicle dynamics are introduced [13]. Finally, we present performance in the two simulators focusing on the domain adaptation problem.

### 1.1. Related Work

The first RL approaches in autonomous vehicles were focused on maneuvers execution [14]. Learning from environment data, these agents directly provide the steering and braking control inputs for the vehicle to execute driving maneuvers in simple scenarios. However, traditionally classic algorithms have been used for low-level control, obtaining good performance [15,16], while RL approaches have slower learning rates and poorer performance. Therefore, we propose to use RL in high-level decision making in much more complex scenarios with multiple adversarial vehicles.

Separating the planning task into high-level decision making and maneuver execution [17] is a way of handling this complexity. An execution layer is in charge of the motion, while a decision-making layer executes the high-level actions. This is known as the hierarchical approach [18].

A second approach that can directly infer control commands from an input obtained directly from sensors is end-to-end learning [9,10]. With the last year’s improvements in deep learning algorithms and the modern hardware computation capabilities, researchers at NVIDIA proposed an end-to-end supervised learning method based on CNN capable of directly steering an automobile [19]. From this pioneering work, many others have used this approach [20,21].

In particular, we review the high-level decision making RL-based approaches at intersections. A Deep Q-Network (DQN) algorithm to deal with intersections is commonly used [22]. In addition, in [23,24], a DQN algorithm to deal with different driver behaviors is used. Some works such as [25,26] train at occluded intersections using a risk-based reward function. The input vector contains information about the ego vehicle, the existing lanes, and the surrounding vehicles. Others, such as [27], are focused on the intentions of the adversarial vehicle, where an RL agent based on cooperation with adversarial vehicles in dense traffic is presented. This author also designs a curriculum learning agent using the concept of level-k behavior [28]. Furthermore, a POMDP-based low-level planner is defined in [29], achieving safe planning results with high-commute efficiency at unsignalized intersections in real time. In [30], an invariant environment representation from the perspective of the ego vehicle based on an occupancy approach is proposed. The agents are capable of performing successfully in unseen scenarios. An additional study with a simple occlusion model is described in this work, where the simulation environment is generated using SUMO. There are also some studies where the state vector is obtained directly from sensor data, such as [3], where a monocular camera is used.

Since our architecture is trained using a curriculum reinforcement learning approach, we present a review of this topic. This process trains first a model with easy tasks and then gradually increases the difficulty of the tasks, improving the performance of the model. A survey of CL can be found in [31,32]. In [4], an automatically generated curriculum-based reinforcement learning for autonomous vehicles in an urban environment is proposed and demonstrates a significant reduction of training time, achieving better performance than traditional methods in intersection traversing and approaching scenarios. Finally, in [11], an end-to-end competitive driving policy for the CARLA simulator is implemented.

### 1.2. Contribution

This paper focuses on developing a safe high-level decision-making framework for different types of intersections. A training environment in which the RL agent will be trained under different situations is defined for this purpose. More specifically, in this work, we present the following achievements:An RL agent capable of learning the desired policy for three types of intersections. Most approaches in the literature use RL to estimate the intentions of the adversarial vehicles, and the agent uses this information to cross an intersection. In our proposal, without any prior information about the scenario, our agent infers not only the intentions of the adversarial vehicles but also the type of intersection.A PPO algorithm with a preprocessing layer for the tactical level. We compare the training performance and the results obtained for both architectures with and without the features extractor. This module is introduced using a hybrid AD architecture, and the whole architecture is validated in a realistic simulator, which includes vehicle dynamics. This approach is an alternative to the trendy end-to-end approaches, showing good results in complex scenarios where end-to-end options do not converge.A novel two-stage training method. We use a model trained in a light simulator as a prior model for training in a realistic simulator. This allows convergence in CARLA within a reasonable training time.

## 2. Hybrid Architecture

This section addresses a completely autonomous driving system, including high-level decision making and maneuver control in a hybrid architecture. It is divided into three levels, as is described in Figure 1. The strategy level defines the trajectory; the tactical level is in charge of the high-level decisions, and the operative level executes the control maneuvers. This paper is focused on the tactical level.

### 2.1. Strategy Level

The path-planning module is described in our previous paper [33]. This module first generates a directed graph of roads and lanes using the HD map input. Then, it calculates a topological (road-lane) route between the ego vehicle location, which is provided by the simulator, and the goal location, applying the Dijkstra algorithm to the previously generated graph. Finally, the route is calculated as a list of way-points centered in the lanes and separated by a given distance. The route is calculated and published in a Robot Operating System (ROS) [34] topic every time a new goal location is set. The road information, such as velocity limit, road lanes, or road type, is also defined in this ROS message.

### 2.2. Tactical Level

High-level decisions are executed at the tactical level, where the simulator information is processed to define the state vector. The RL agent executes an action at every time step. This level will be explained in detail in Section 4.

### 2.3. Operative Level

The maneuver execution is carried out by a classic controller [35] that performs a smooth interpolation of the way-points given by the planner. Before the navigation starts, a velocity profile is generated using the curvature radius of each trajectory section and the velocity limit defined for these sections. During the navigation, the velocity command is adjusted using this profile, and the steering command is set using Linear Quadratic Regulator (LQR) techniques to ensure trajectory tracking and a smooth lane change execution.

### 2.4. Perception

The perception data are directly obtained from the simulator’s ground truth. In real applications, these data could be obtained using the fusion of cameras and lidars.

## 3. Background

A Markov decision process (MDP) is defined by a tuple (S,A,P,R) in which *S* is a set of states, *A* is a set of actions, *P* is the probability that action *a* in state *s* at time *t* will lead to state $s^{\prime }$ at time t+1 and *R* is the reward function. An agent in state s∈S takes an action a∈A transitioning to $s^{\prime }$ and receiving a reward $R(s,a,s^{\prime })$, as shown in Figure 2. In an MDP, the agent receives a $s^{\prime }$ directly. The optimal policy $\pi^{*}\left(s\right)$ maps a state to action.

In this work, we propose DRL to solve MDPs. Our approach focuses on a policy-based method, which directly learns the policy function that maps state to action. These methods work by computing an estimator of the policy gradient and plugging it into a stochastic gradient ascent algorithm. The loss function for updating an RL policy has the general form:(1)$$L^{PG}\left(\theta \right)={\hat{E}}_{t}\left[log\pi_{\theta }\left(a_{t}|s_{t}\right){\hat{A}}_{t}\right]$$
where $E_{t}$ is the expectation, $\pi_{\theta }$ is the policy and ${\hat{A}}_{t}$ is an estimator of the advantage function at a time step *t*.

While it is appealing to perform multiple steps of optimization on this loss function $L^{PG}\left(\theta \right)$, doing so is not well-justified, and empirically, it often leads to destructively large policy updates.

In particular, we use the Proximal Policy Optimization (PPO) algorithm [36], which proposes a clipped surrogate loss function and combines the policy surrogate and a value function error term:(2)$$L_{t}^{CLIP+VF+S}\left(\theta \right)={\hat{E}}_{t}\left[L_{t}^{CLIP}\left(\theta \right)-c_{1}L_{t}^{VF}\left(\theta \right)+c_{2}S\left[\pi_{\theta }\right]\left(s_{t}\right)\right]$$
where $c_{1}$, $c_{2}$ are coefficients, *S* denotes an entropy bonus, and $L_{t}^{VF}\left(\theta \right)$ is a squared-error loss ${\left(V_{\theta }\left(s_{t}\right)-Vt^{targ}\right)}^{2}$. $L_{t}^{CLIP}$ is the clipped surrogate objective, which takes the form:(3)$$L_{t}^{CLIP}\left(\theta \right)={\hat{E}}_{t}\left[min\left(r_{t}\left(\theta \right){\hat{A}}_{t},clip\left(r_{t}\left(\theta \right),1-\epsilon ,1+\epsilon \right){\hat{A}}_{t}\right)\right]$$
where epsilon is a hyperparameter and $r_{t}\left(\theta \right)$ is the probability ratio:(4)$$r_{t}\left(\theta \right)=\pi_{\theta }\left(a_{t}|s_{t}\right)/\pi_{\theta_{old}}\left(a_{t}|s_{t}\right)$$

The value of $L_{t}^{CLIP}$ is the minimum of the clipped and the unclipped objective, so the final objective is a lower bound on the unclipped objective. Large policy updates are avoided by clipping this probability ratio inside the interval [1−ϵ,1+ϵ].

The PPO algorithm runs the policy for *T* timesteps within a given length *T* trajectory segment and uses the collected samples for an update. A truncated version of generalized advantage estimation is the choice taken, which has the form: (5)$$\begin{array}{c} {\hat{A}}_{t}=\delta_{t}+\left(\gamma \lambda \right)\delta_{t+1}+\cdots +\cdots +{\left(\gamma \lambda \right)}^{T-t+1}\delta_{T-1}\\ where\;\delta_{t}=r_{t}+\gamma V\left(s_{t}+1\right)-V\left(s_{t}\right) \end{array}$$

In each iteration, each of *N* (parallel) actors collects *T* timesteps of data. The surrogate loss function is constructed on these NT timesteps of data and is optimized with minibatch SGD for K epochs.

## 4. Reinforcement Learning at Intersections

This section describes the application of RL techniques to realistic environments. First, we define the representation of the intersection scenarios; then, we explain the possible scenarios the vehicle will find at intersections; and finally, the neural network structure and the features extractor are described.

### 4.1. Modeling Intersections

Our goal is to use an RL agent to execute high-level decisions at intersection scenarios, dealing with a continuous state vector and a set of discrete actions. For this purpose, we define the intersection scenarios as an MDP.

#### 4.1.1. State

The state of a vehicle, as shown in Figure 3, is defined by its distance to the intersection point and its longitudinal velocity: $s_{i}=\left\{d_{i},v_{i}\right\}$. These values are normalized between [0,1]. We define the state vector as the collection of the individual states of the two closest vehicles for each lane and the ego vehicle.
(6)$$s=\{d_{ego},v_{ego},d_{adv1},v_{adv1},d_{adv2},v_{adv2},d_{adv3},v_{adv3},d_{adv4},v_{adv4}\}$$

#### 4.1.2. Action

We propose a discrete action space formed by just two high-level actions. The low-level controller is in charge of performing smooth driving based on these actions. These actions are focused on when the ego vehicle has to cross the intersection and when it has to stop. The action space is defined as: a={stop,drive}. Both actions set a desired velocity, *stop* sets 0 m/s and *drive* sets the nominal velocity of 5 m/s.

#### 4.1.3. Reward

An RL algorithm aims to maximize the expectation of the discounted future reward. This means that different reward functions will lead to different optimal policies. Our goal is to drive through the intersection as fast as possible, avoiding the adversarial vehicles, being the reward function defined in terms of success or failure. A negative reward is given when a collision takes place, and a positive reward is given when the vehicle reaches the success point once it has crossed the intersection. To encourage the ego vehicle to move, we propose a cumulative reward based on its longitudinal velocity. In addition, when an episode ends, a negative reward is given relative to its duration, where *t* is the training episode duration and $t_{out}$ is the timeout time. The reward function consists of the following terms:Reward based on the velocity: $k_{v}*v_{ego}$;Reward for crossing the intersection: +1;Penalty for collisions: −2;Penalty relative to the episode duration: $-0.2t/t_{out}$.

Large reward values can lead to large gradients according to [37]. Therefore, all the rewards are scaled to keep the total reward $r_{t}\in \left[-1,1.2\right]$.

### 4.2. Intersection Scenarios

We present the hypothesis that the RL agent (ego vehicle) is capable of driving only based on the position and speed of the adversarial vehicles (adversaries) since the type of intersection is not given in the state vector. To prove this, we have designed three different scenarios, all of them with the same dimensions but with different features: traffic light intersection, stop sign intersection and uncontrolled intersection.

The complexity of managing regulatory elements, such as a stop signal or a traffic light, is solved using classic approaches in the decision-making module. We implement a simple high-level module for this purpose in order to obey these traffic rules. As a consequence, we evaluate the ego vehicle without this module. The RL agent should infer the type of intersection and adapt its behavior only based on the information of the adversaries. We just focus on crossing the intersection in the fastest way ensuring collision avoidance.

The definition and the implementation of these scenarios are completed in two simulators, first in SUMO, which does not consider vehicle dynamics and in CARLA, in which vehicle dynamics are included. In this section, we focus on the definition of the physical environment and the behavior of the vehicles. Traffic generation is one of the pillars to obtain a realistic simulation.

In SUMO, the traffic density and vehicle features are similar for the three scenarios. A new vehicle is generated in the simulation every sec∈[5,10] s. These vehicles are 4 m long and follow a standard Intelligent Driver Model (IDM) movement [38] with a target velocity of v∈[4,6] m/s. The three scenarios are defined by two roads with two lanes each. The length of these roads is 200 m, and they are perpendicular to the intersection point in the middle.

In CARLA, six vehicles are spawned in random locations close to the intersection. These vehicles start moving when the ego vehicle approaches. They are commanded by the CARLA Autopilot, which has different driving modes defined by the CARLA Traffic Manager. All these vehicles have the dimensions of a Tesla Model 3 and a target velocity v∈[4,6] m/s. These roads are 50 m long, and they are perpendicular to the intersection point in the middle.

We need to define the scenarios in the two simulators slightly different to obtain similar intersection scenarios. In CARLA, the simulation is slower than in SUMO, so we use a shorter road and generate fewer adversaries, but the ego vehicle faces a similar situation when it approaches the intersection.

#### 4.2.1. Uncontrolled Intersection

This is the most difficult scenario, since the ego vehicle has to deal with many different behaviors by the adversaries. Their behavior may vary depending on the ego vehicle and the other adversaries, since they do not follow signal rules. The ego vehicle and the adversaries all continue straight when crossing the intersection. This route, described in Figure 4a, is the same for the three intersection scenarios.

In the absence of a signal that regulates the right of way, a driver is obliged to yield to vehicles approaching from his right. In this scenario, the adversaries tend to follow this rule, but in some cases, this may not happen.

#### 4.2.2. Traffic Light Intersection

Solving controlled intersections may seem a deterministic problem just by following the signals, but in autonomous driving, this is not the case. The task of determining the state of the traffic light is quite hard due to the huge variety of these in different countries.

We propose a novel approach to this problem, learning the desired behavior based on the other vehicles. In this way, collisions can be avoided even though the ego vehicle misses the traffic light. For this purpose, we set four traffic lights at the intersection, one for each lane. A predefined sequence for the simulation is defined in Table 1.

Each episode starts at a random phase p∈[1,4]. The adversaries follow the traffic light rules and do not yield if the light is green.

In Figure 4b, the traffic light scenario is represented. Following the SUMO notation, the four traffic lights states are described as GrGr, where the traffic lights are green “*G*” and red “*r*”, alternatively.

#### 4.2.3. Stop Sign Intersection

The complexity of this scenario is that the ego vehicle must learn to yield. As described in Figure 4c, the ego vehicle has a stop signal in its lane while the adversaries can freely cross the intersection. These adversaries never stop, forcing the ego vehicle to stop and cross when there is a big enough gap.

### 4.3. Neural Network Architecture

We propose a features extractor module and an actor–critic approach for the RL agent. The algorithm structure is shown in Figure 5.

The state vector is defined by the distances and velocities of the vehicles in the scenario as described in Section 4.1.1. Both adversaries and ego vehicle features are introduced in the extractor separately and concatenated in a single vector. The features extractor is shared between the actor/critic network being tuned in an unsupervised way. The concatenated vector is the PPO input. This algorithm has two models, the actor and the critic. The actor chooses the action and corresponds with the policy, while the critic corresponds with the value function. Both models are defined with a simple dense multi-layer perceptron (MLP) formed by two hidden layers of 128 neurons each. By keeping the network simple, we prevent overfitting and achieve faster training.

## 5. Experiments

In this section, we present some details about the training methodology, the evaluation metrics and the obtained results.

### 5.1. Training the Algorithm

We present a two-stage training method. We propose a first training stage, with a large number of episodes in a simple simulator. This training stage has two phases. In the first phase, adversaries are only generated in only one lane, and in the second phase, adversaries are generated in both lanes. The second training stage, with is completed with the prior model, is in charge of adapting this model to a realistic simulator. The training methodology is described in Figure 6.

In the first training stage, velocity commands and maneuvers are defined by SUMO, while in this second stage, this is performed by CARLA. The communication between the different levels (strategy, tactical and operative) is carried out by ROS.

#### 5.1.1. SUMO Simulator

SUMO is a simulation tool that allows the creation of driving scenarios and access to the environment information. It does not provide sensor implementation or vehicle dynamics, while it focuses on driving and decisions, which allows a really fast simulation. We have designed three intersections as described in Section 4.2. This simulator has been developed using the OpenAI Gym [39] library, which allows an easy way of creating custom environments for RL applications and the traci tool [40], which provides access to the simulation information.

The episode starts with the ego vehicle at the beginning of the road, and a traffic flow is generated in each lane. For each step of the simulation, a state vector is calculated, the RL agent executes an action and a reward is calculated for that action at that moment. The episode ends when the ego vehicle collides, reaches the end of the road or there is a timeout.

We propose a curriculum learning approach by adding complexity to the scenario. Training the RL algorithm under high-density traffic intersections does not converge to an optimal policy. The policy obtained avoids collisions but does not cross the intersection; thus, we propose a first training phase only with vehicles driving in one of the intersecting lanes and a second training phase with vehicles in both lanes. The algorithm is trained over 5M episodes for each phase.

#### 5.1.2. CARLA Simulator

SUMO is an autonomous driving simulator that emulates real driving but does not simulate the dynamics of the vehicle. To add complexity to this study, we propose extrapolating the models obtained in this simulator to the realistic CARLA simulator, which is developed to design and validate autonomous driving systems.

One of the most challenging tasks in RL is domain adaptation. We maintain the essential inputs and outputs, so the content does not vary for the agent. The intersection scenarios of the two simulators are shown in Figure 7.

In CARLA, three vehicles are spawned in each lane. These have different behaviors: they can yield, cross or stop randomly. They are commanded by the CARLA autopilot. As in SUMO, each episode ends when the collision sensor is activated, if the ego vehicle reaches the end of the road or if the duration of the episode exceeds the timeout value. During these episodes, when the tactical level requires an action, this is sent via ROS to the operative level.

### 5.2. Evaluation Metrics

To quantify the performance of our system, we evaluate the ego vehicle in terms of success and simulation time. These metrics are defined as:$success\;\left(\%\right)=end\;reached/n_{e}$$t_{avg}=\sum t_{n}\;/\;n_{e}$where the number of episodes $n_{e}$ is defined for each evaluation, the simulation time is measured in seconds and the commanded velocity is 5 m/s.

It should be pointed out that the road in the SUMO simulator is significantly longer than the road in the CARLA simulator. Because of this, the average time in SUMO is expected to be longer than in CARLA.

We define four scenarios for these experiments, one for each type of intersection defined in Section 4.2 and one combining all of them in a random way. The adversaries are generated in the same way that they were generated during the training. The previously trained algorithms are tested in this validation environment.

First, we focus on SUMO, running 100 test episodes in each scenario for each algorithm trained: one phase training PPO (1-PPO), two-phase training PPO (2-PPO), one-phase training PPO with features extractor (1-FEPPO) and two-phase training PPO with features extractor (2-FEPPO).

The model with better results in SUMO is used as prior data to conduct the second training stage in CARLA, using both the network and the weights. This prior model (SUMO) and the final model (CARLA) are compared in terms of success and simulation time in all four scenarios in CARLA. We run 100 test episodes for each scenario to obtain the results presented in the following section.

### 5.3. Results

In this section, we describe the results obtained during this research. We define four scenarios and expose the performance of the different algorithms in terms of success and average episode time. In addition, we compare our architecture with other similar architectures in the state of the art. The GPU used for all the experiments described below is the NVIDIA Titan with 12 GB of VRam.

In Table 2, we present the mean values along 100 test episodes in the SUMO simulator for the previously described algorithms and scenarios. The 2-FEPPO obtains the best results with no collisions in the traffic light scenario and a 95% success in the combined scenario.

In Table 3, we present the mean values along 100 test episodes in the CARLA simulator for the previously described algorithms and scenarios. The results obtained with the 2-FEPPO are worse in CARLA than in SUMO. After the second training stage, a 78% success rate in the combined test scenarios is obtained by the RL agent.

It is a difficult task to present a comparison with the state of the art, since different scenarios and problems are defined in different works. However, in Table 4, we present a comparison in terms of success rate at an uncontrolled intersection. This table shows the results of our approach and some other similar approaches that can be found in the literature. The obtained success rates indicate that our results are on par or even better than other competitive approaches.

In addition, we study the influence of virtual sensors in the RL agent regarding an agent trained using only ground truth data. For this experiment, we have not added another training stage, so we can see the effects of the uncertainty produced by the sensor’s detection. We present a first approach where the state vector is directly obtained from the SUMO simulator and a second more realistic approach, where this vector is defined using an estimation of the parameters. We have studied this estimation in a 3D object detection benchmark (KITTI dataset [41]) where ground truth is provided by a Velodyne laser scanner and GPS localization system. We have processed these sensor data using the PointPillar algorithm [42], and we have obtained a maximum distance error of ±5 m (for distances between 0 and 100 m). Finally, we have modified our state vector by adding this error. In Table 5, we present the evaluation metrics of these two approaches. It can be seen that the results are better using ideal ground truth data obtained directly from the simulator (distance error equals 0). However, very good results are also obtained working with data simulating sensor errors (distance error equals 5%).

These results can be replicated through our repository found in the Appendix A.

### 5.4. Discussion

RL is defined as a trial and error process, where the experience of modeling the environment and the reward is the key to success. Our first approach was training the agent from scratch using the CARLA simulator, but we discarded training in this way due to the large episodes required for convergence. Then, we used the SUMO simulator, where we were able to train the agent during 30 k episodes in a few hours. Using SUMO training as prior in CARLA training took 10.5 h. In Table 6, a comparison between the training time for each simulation is shown.

Training the ego vehicle in dense traffic intersections using the PPO algorithm did not converge into an optimal solution: the ego vehicle did not cross the intersection but avoided collisions. With this behavior, the mean value of the total rewards was lower than 1. We solved this by a curriculum learning approach adding vehicles in different phases. By this, we obtained an optimal policy: the ego vehicle was able to cross the intersection avoiding adversarial vehicles and reaching a mean value for the total rewards close to 1.2. This is the maximum value of the total reward and guarantees that the chosen policy is optimal. Then, we presented the features extractor architecture, which led us to obtain better results in performance and convergence. The two-phase trained algorithms obtained higher success percentages, the 2-FEPPO of them being the best. The one-phase trained algorithms were not trained in dense traffic scenarios, so a high number of failures was expected. Regarding the intersection scenarios, all vehicles obtained the best results in the traffic light scenario and the worst metrics in the stop scenario. This makes sense considering that adversaries do not yield when there is a stop sign for the ego vehicle, but they do yield when the traffic light is red for them. Considering that the success rate for the proposed architecture is quite good, we observed that the ego vehicle did not always follow traffic rules. If it finds a gap when the traffic light is red, it will pass. The ego vehicle drives only according to the adversary’s behavior, so a simple higher layer would be necessary to satisfy the traffic rules.

Figure 8 shows a visualization of the RL agent at the traffic light intersection, with the time, the actions and the ego vehicle velocity data information. The ego vehicle stops when the adversary is getting close to the intersection; when this adversary goes through the intersection, the ego vehicle takes the action to drive and cross the intersection.

In addition, we analyze the performance of adding a features extractor between the state vector and the PPO input. In Figure 9, we show the mean reward for the episodes along the training phase. It is clear that training with the features extractor layer converges faster and even the mean reward is higher. It should be noted that the second phase of the FEPPO converges in around 1 M episodes.

The transfer of knowledge is a difficult task, since the two environments have some differences. Our priority is to set the state vector and the transition function of both simulators as similar as possible, but they are never the same. As mentioned in previous sections, a second training stage is performed using the 2-FEPPO model weights.

The resulting algorithm of this second stage does not obtain the best policy in the CARLA simulator. The main difference between the two simulators is in the vehicle’s dynamics. When an action is taken in SUMO, this action happens immediately, but in CARLA, a response time is added. Adding dynamics to the vehicles’ movements adds complexity to the intersection problem. The resulting behavior is that the ego vehicle crosses the intersection consistently, but sometimes it collides with other vehicles.

A visualization of the RL agent at an uncontrolled intersection is shown in Figure 10 along with the time, the actions and the ego vehicle velocity data information. The ego vehicle stops when the adversary is getting close to the intersection, and as it goes away, the ego vehicle drives and crosses the intersection.

## 6. Conclusions and Future Works

The results of this paper state that a Proximal Policy Optimization algorithm can execute decisions at intersection scenarios.

The algorithm learns the policy with different types of intersections with no prior information about the type of scenario but only the relative distance and velocity of the surrounding vehicles. The ego vehicle is able to drive in complex scenarios, but for inferring traffic rules from other participants, it would be necessary to have a reward focused on this feature.

Furthermore, the application of curriculum learning techniques for the RL agent training allows improvements in the metrics obtained and faster convergence. The implementation of a features extractor module not only improves the training phase but obtains better metrics during the evaluation tests.

In the near future, we plan to extend this research by obtaining the state vector from real sensor data instead of obtaining them directly from the ground truth. The main advantage of the proposed architecture is that it is ROS based, which allows a fast and easy integration in our real vehicle [43]. We have already conducted some experiments in a real environment on our university campus, using a Petri nets-based decision-making module. Our future goal is to replace this module with the system described in this work and compare the results.

## Figures and Tables

**Figure 1 sensors-22-08373-f001:**
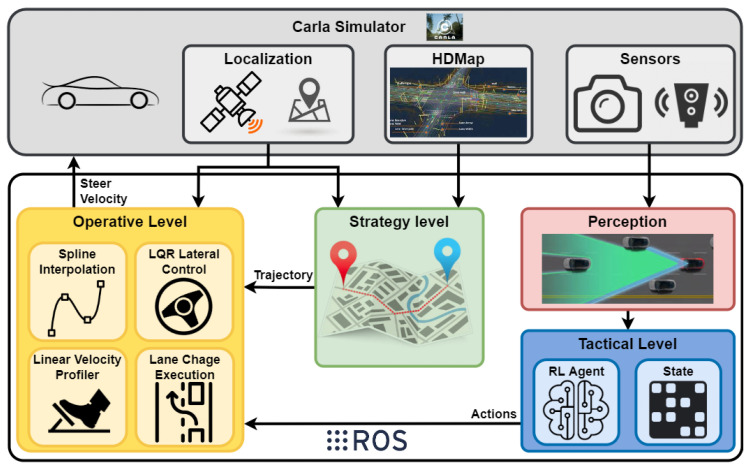
The proposed hybrid architecture. The strategy level defines a trajectory with the map information and the ego vehicle location. The tactical level executes high-level actions in correlation with the perception information. The operative level combines the trajectory and the actions, calculating the driving commands.

**Figure 2 sensors-22-08373-f002:**
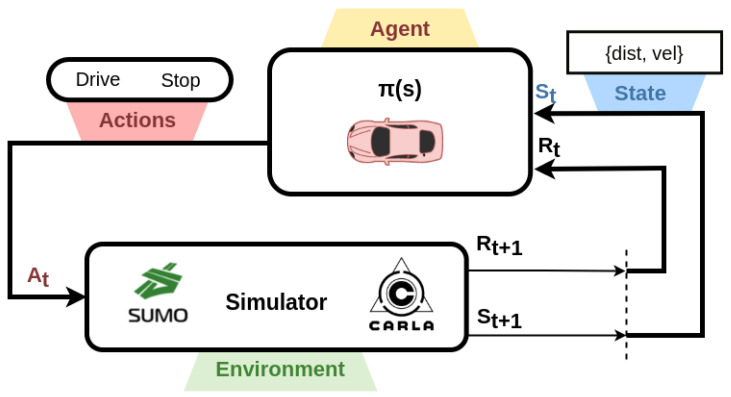
The state vector is extracted from the simulator. The agent selects an action considering this matrix and the reward for this action.

**Figure 3 sensors-22-08373-f003:**
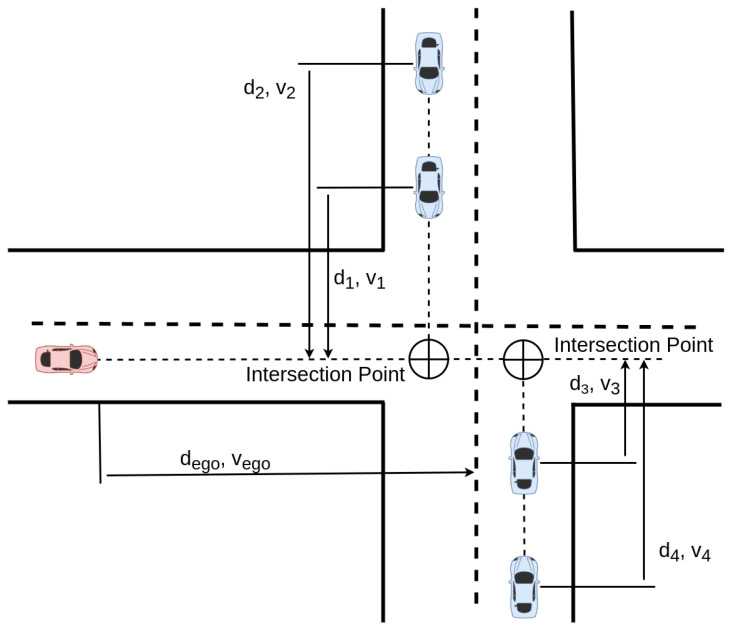
The ego vehicle (red) and the adversary (blue) are represented. The velocity of each vehicle and the distances to the intersection point are represented. This point is defined by the intersection of the trajectories of the vehicles.

**Figure 4 sensors-22-08373-f004:**
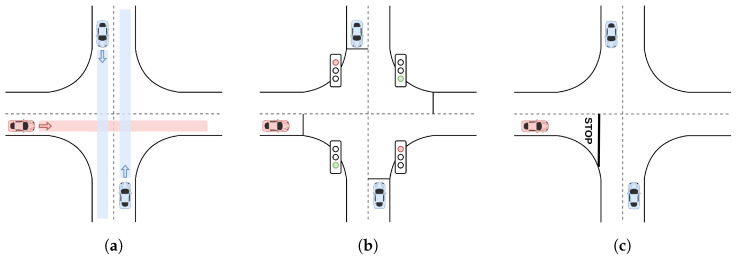
Definition of the three intersection scenarios. (**a**) Uncontrolled intersection scenario: The ego vehicle (red) follows the red trajectory and the adversaries (blue) follow the blue trajectory. (**b**) Traffic light intersection scenario: The ego vehicle (red) has a green light while the adversaries (blue) have a red light. (**c**) Stop intersection scenario: The ego vehicle (red) has a stop signal while the adversaries (blue) can freely cross the intersection.

**Figure 5 sensors-22-08373-f005:**
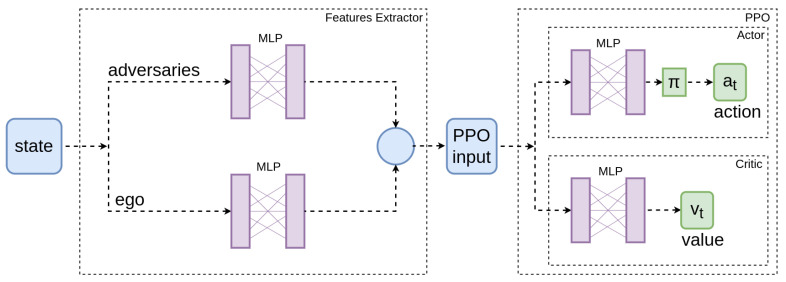
Neural network: After two fully connected layers, both adversaries and ego vehicle features are concatenated and fed the actor–critic structure with two layers of 128 neurons each.

**Figure 6 sensors-22-08373-f006:**
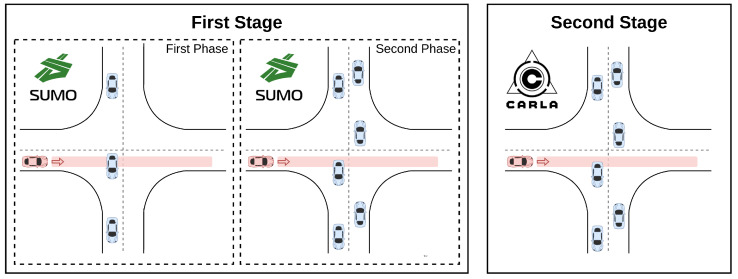
Training methodology: The first stage in SUMO with two phases of increasing complexity and the second stage in CARLA.

**Figure 7 sensors-22-08373-f007:**
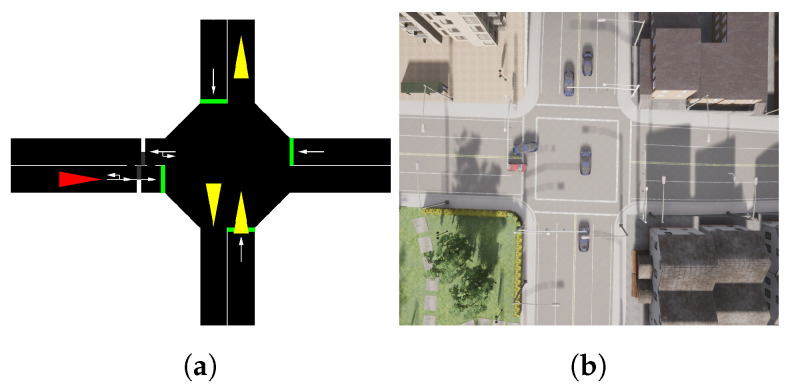
CARLA and SUMO intersection scenarios. (**a**) Bird-eye-view of the intersection in SUMO Simulator. (**b**) Bird-eye-view of the intersection in CARLA Simulator.

**Figure 8 sensors-22-08373-f008:**

RL agent (red) at the intersection with the traffic light. Velocity (v), action (a) and time (t) are described for each frame. (**a**) v = 5; a = drive; t = 9.5 (**b**) v = 0; a = stop; t = 13.7 (**c**) v = 2; a = drive; t = 16.2 (**d**) v = 5; a = drive; t = 22.5.

**Figure 9 sensors-22-08373-f009:**
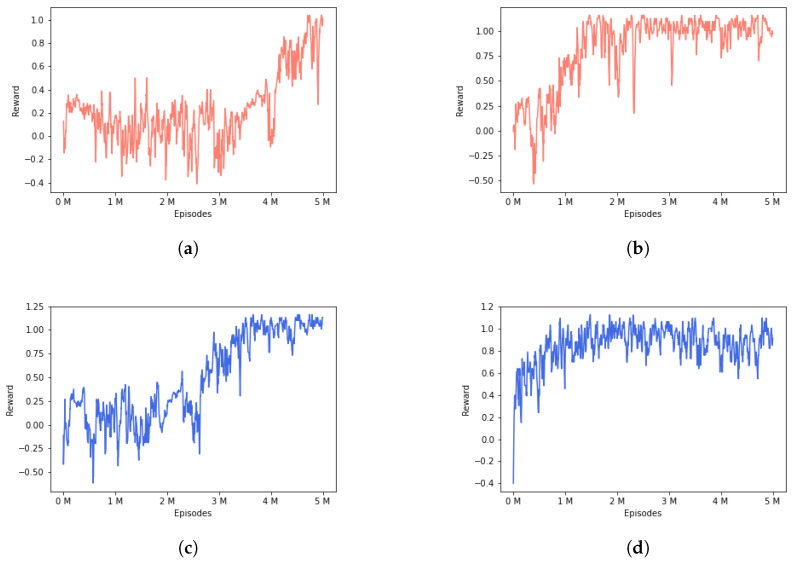
Episodes mean rewards. Comparison between the two approaches described in the paper: with (**right**) and without (**left**) features extractor module. The red charts correspond to the first training phase and the blue charts corresponds to the second training phase. (**a**) One-phase PPO episodes mean reward with adversaries in one lane. No features extractor layer. (**b**) One-phase FEPPO episodes mean reward with adversaries in one lane. Features extractor layer. (**c**) Two-phase PPO episodes mean reward with adversaries in two lanes. No features extractor layer. (**d**) Two-phase FEPPO episodes mean reward with adversaries in two lanes. Features extractor layer.

**Figure 10 sensors-22-08373-f010:**
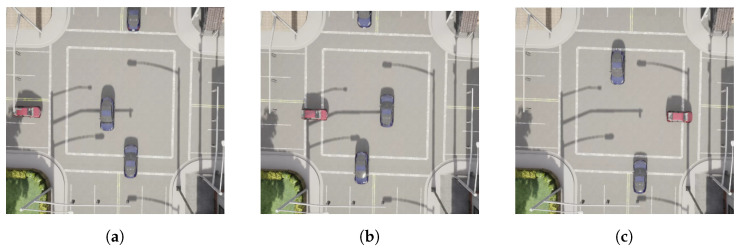
RL agent (red) at uncontrolled intersection. Velocity (v), action (a) and time (t) are described for each frame. (**a**) v = 3; a = stop; t = 8.5 (**b**) v = 4; a = drive; t = 12.7 (**c**) v = 5; a = drive; t = 17.2.

**Table 1 sensors-22-08373-t001:** Traffic Lights Sequence.

Phase	Duration (s)	State
1	20	GrGr
2	2	yryr
3	20	rGrG
4	2	ryry

**Table 2 sensors-22-08373-t002:** Evaluation metrics results in SUMO simulator: comparison between algorithms trained with and without features structure.

	Traffic Light		Stop Signal		Uncontrolled		Combination	
	success	$t_{\mathrm{avg}}$	success	$t_{\mathrm{avg}}$	success	$t_{\mathrm{avg}}$	success	$t_{\mathrm{avg}}$
	**(%)**	**(s)**	**(%)**	**(s)**	**(%)**	**(s)**	**(%)**	**(s)**
1-PPO	53	112	37	93	23	109	30	105
2-PPO	95	67	78	78	87	63	88	71
1-FEPPO	61	104	48	82	30	111	37	102
2-FEPPO	100	43	90	94	95	55	95	85

**Table 3 sensors-22-08373-t003:** Evaluation metrics results in CARLA simulator: comparison between the model trained in SUMO (2-FEPPO) and the model trained in CARLA (SUMO + CARLA).

	Traffic Light		Stop Signal		Uncontrolled		Combination	
	success	$t_{\mathrm{avg}}$	success	$t_{\mathrm{avg}}$	success	$t_{\mathrm{avg}}$	success	$t_{\mathrm{avg}}$
	**(%)**	**(s)**	**(%)**	**(s)**	**(%)**	**(s)**	**(%)**	**(s)**
SUMO	78	17	35	19	47	19	50	19
CARLA	83	17	70	19	75	16	78	17

**Table 4 sensors-22-08373-t004:** Comparison of success rate between different approaches.

Architecture	Success Rate (%)
2-FEPPO	95
MPC Agent [24]	95.2
Level-k Agent [28]	93.8
Sc04 Left Turn [30]	90.3

**Table 5 sensors-22-08373-t005:** Evaluation metrics results in SUMO simulator: ground truth vs. sensor data simulation.

	Traffic Light		Stop Signal		Uncontrolled		Combination	
	success	$t_{\mathrm{avg}}$	success	$t_{\mathrm{avg}}$	success	$t_{\mathrm{avg}}$	success	$t_{\mathrm{avg}}$
	**(%)**	**(s)**	**(%)**	**(s)**	**(%)**	**(s)**	**(%)**	**(s)**
Ground Truth	100	43	90	94	95	55	95	85
Sensor Data	96	42	88	92	94	51	91	80

**Table 6 sensors-22-08373-t006:** Simulation time.

Simulator	No. of Episodes	Time (h)
SUMO	30 k	5
CARLA (estimated)	30 k	1650
SUMO + CARLA	30 k + 1 k	10.5

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
