# Peer review of "Reinforcement Learning-Based Autonomous Driving at Intersections in CARLA Simulator"

_sensors, 2022, doi:10.3390/s22218373_

Round 1

Reviewer 1 Report

This work addresses the behavior of self-driving at intersections, using deep reinforcement learning combined with curriculum learning, 3 scenarios are configured to test the proposal.

·        the performance of an RL algorithm depends on the definition of the reward function, it would be interesting if the authors detailed in greater depth how they decided to define their reward function in this way

·        How is it guaranteed that the chosen stock policy is optimal and not suboptimal?

·        authors are advised to check for spelling errors in the text

·        The references used are current and significant, however, it is recommended to add the gab offered by each reference in relation to the problem to be solved.

·        The results presented are based on simulations, however, it would be interesting to add experimental results, what did the authors need to achieve this?

Author Response

Response to Reviewer 1 Comments

Dear Reviewer:

Thank you very much for your review of our paper. We sincerely appreciate all valuable comments and suggestions, which helped us to improve the quality of the article. A major revision of the paper has been carried out to take all of the corrections into consideration. Specific responses to the reviewer's comments are provided below:

Point 1:  The performance of an RL algorithm depends on the definition of the reward function, it would be interesting if the authors detailed in greater depth how they decided to define their reward function in this way.

Response 1: An RL algorithm aims to maximize the expectation of the discounted future reward. This means that different reward functions will lead to different optimal policies. Our goal is to drive through the intersection as fast as possible avoiding adversarial vehicles. To drive as fast as possible, we define a positive reward that depends on the ego vehicle velocity and a negative reward that depends on the episode time. We also define a negative reward based on collisions to avoid adversarial vehicles. Finally, we define a positive reward for the vehicle when it  crosses the intersection successfully. Large reward values can lead to large gradients. Therefore, all the rewards are scaled to keep the total reward rt ∈ [−1, 1.2].

We have extended Section 4.1.3 by including an explanation of how we decided to define the reward function.

Point 2: How is it guaranteed that the chosen stock policy is optimal and not suboptimal?

Response 2: The total reward of an episode has a maximum value of 1.2. The value function is the expectation of this total reward. The optimal policy is the policy that results in the optimal value function. We can conclude that we have achieved an optimal policy considering that the total reward of the RL agent is almost the maximum value. Figure 9 shows the mean value evolution of the total reward.

We have extended Section 5.4 describing how we first obtained a suboptimal policy and how then we obtained an optimal one using a curricular learning approach (lines 395-400).

Point 3: authors are advised to check for spelling errors in the text.

Response 3: To improve the quality of our paper, we have used the “Grammarly” tool, which is a writing assistant that reviews spelling, grammar, punctuation and clarity of English text. With this tool, we have checked for spelling errors. In addition, we have sent the paper to english-speaking proofreader to improve the English. There are modifications all over the text that can be tracked using the review tab in Overleaf tool.

Point 4: The references used are current and significant, however, it is recommended to add the gap offered by each reference in relation to the problem to be solved.

Response 4: We have done an extension of the references in Section 1.1.

Line 74 - " Also, in [ 23, 24 ], a DQN algorithm to deal with different drivers behaviours is used. "

Line 77 - Others, like [ 27 ] are focused on the adversarial vehicle's intentions, where an RL agent based in cooperation with adversarial vehicles in dense traffic is presented. This author, also designs a curriculum learning agent using the concept of level-k behaviour [28].

Line 82 - Also, in [30], an invariant environment representation from the perspective of the ego vehicle based on an occupancy approach is proposed. The agents are capable of performing successfully in unseen scenarios An additional study with a simple occlusion model is described in this work, where the simulation environment is generated using SUMO.

Point 5: The results presented are based on simulations, however, it would be interesting to add experimental results, what did the authors need to achieve this?.

Response 5: We first present this work in simulation to evaluate our architecture. The next step for this research is to use the sensors provided by the simulator, instead of the ground truth, to generate the state vector. With this new state vector, we will study the robustness of the proposed solution. Once we verify our architecture in a realistic simulation, not only with realistic dynamics but also with realistic sensors; we will be able to implement the architecture in the real vehicle. We have added a study about this problem. To do this, we have studied this error in a 3D object detection benchmark (KITTI dataset) where ground-truth is provided by a velodyne laser scanner and GPS localization system. We have procesed this data using the PointPillar algorithm and we have obtained a maximum distance error of ±5 m (for distances between 0 and 100m). Finally, we have modified the state vector by adding this error. The results of this experiment are shown in Table 5 and a discussion about these results can be found in Section 5.3.

The main advantage of the proposed architecture is that it is ROS based, which allows a fast and easy integration in our real vehicle. We have already done some experiments in a real environment on our university campus. We have already validated some modules in real environments. In our research group, we have evaluated the operative level (Waypoint Tracking Controller), a Petri Nets based Decision Making (DM) module, and some perception modules in the real platform. It will be possible to integrate the proposed work in the near future. Our goal is to replace the classic DM module with the one proposed in this work. The main difficulty for our research group is in the implementation of the real scenarios since we need time and expensive infrastructure. We have improved Section 6 (Future works) by adding some comments about this issue. 

Reviewer 2 Report

In this paper the authors propose a framework based on reinforcement learning for autonomous driving at intersections considered among the situations that cause a large number of accidents. Three types of intersections have been studied: a control, traffic light intersection and stop intersection considered as the most difficult scenario where all other vehicles have priority.

The approach is well detailed and considers solving the problem without prior information on the type of intersection deduced based on the position and speed of the opposing cars. The results show the usefulness of the CL for improving the performance of the model as well as the combination of feature extraction and PPO. The proposed solution has been validated using first sumo for the preparation of the models and later on the carla simulator which takes into consideration dynamic vehicles.  

The work is very interesting and there are some questions that need further clarification:

The speed and distance of the adv vehicles are the basic criteria for the control of the ego vehicle, what about the quality of the estimation of these parameters and how the mis-estimation can impact the robustness of the proposed solution.

In the simulator, the perception of the environment (the intersection) is complete, unless I am mistaken, can the simulator take into consideration the latency of recovery of this information, in which case it would have an impact on the functioning of the solution.

This work is very interesting and the approach is well structured and detailed. The results are very interesting, especially the convergence time of the learning models boosted by the CL approach.

Author Response

Response to Reviewer 2 Comments

Dear Reviewer:

Thank you very much for your review of our paper. We sincerely appreciate all valuable comments and suggestions, which helped us to improve the quality of the article. A major revision of the paper has been carried out to take all of the corrections into consideration. Specific responses to the reviewer's comments are provided below:

Point 1:  The speed and distance of the adv vehicles are the basic criteria for the control of the ego vehicle, what about the quality of the estimation of these parameters and how the mis-estimation can impact the robustness of the proposed solution.

Response 1: In this work, the speed and distance of the adversarial vehicles are directly obtained from the ground truth of the simulator. We assume that there is no error in the estimation of these parameters. However, your comment is really interesting, so we have added a study about this problem. To do this, we have studied this error in a 3D object detection benchmark (KITTI dataset) where ground-truth is provided by a velodyne laser scanner and GPS localization system. We have procesed this data using the PointPillar algorithm and we have obtained a maximum distance error of ±5 m (for distances between 0 and 100m). Finally, we have modified the state vector by adding this error. The results of this experiment are shown in Table 5 and a discussion about these results can be found in Section 5.3.

Point 2: In the simulator, the perception of the environment (the intersection) is complete, unless I am mistaken, can the simulator take into consideration the latency of recovery of this information, in which case it would have an impact on the functioning of the solution.

Response 2: The simulator cannot take into consideration the latency of the recovery of the information. In real applications, the sensor's delay and the data processing time can impact the behaviour of a decision making system. To evaluate how this may modify the performance of our proposed architecture, we have simulated a one timestep delay (100ms) in the state vector. We have verified that this is the processing time of the camera and lidar data in the CARLA simulator. Finally, we cannot observe any relevant changes in the results obtained. So we can conclude that this delay does not significantly affect the agent in urban scenarios at low velocities in simulation.

Reviewer 3 Report

This work proposes a solution to avoid collisions at intersections for autonomous vehicles. 

The results should be compared with the state of the art advantages and limitations highlighted. 

In table 3, it is not clear why the CARLA simulator had lower tavg.  It is also clear that the transfer learning from SUMO to CARLA is not very successful. A deeper discussion is needed why there is this major impact as success falls drastically (compared to table 2). 

Some other minor corrections:

Line 85 - "increases the difficult of the tasks" -> "increases the difficulty of the tasks"

Line 172 - "scenarios the vehicle will found at intersections" -> "scenarios the vehicle will find at intersections"

Line 221 - "These roads length is 200 m and" -> "The length of these roads is 200 m and"

Line 257 - "has a stop signal in his lane while the adversaries" -> "has a stop signal in its lane while the adversaries"

Line 334 - "being used both the network and the weights." -> "using both the network and the weights."

Author Response

Response to Reviewer 3 Comments

Dear Reviewer:

Thank you very much for your review of our paper. We sincerely appreciate all valuable comments and suggestions, which helped us to improve the quality of the article. A major revision of the paper has been carried out to take all of the corrections into consideration. Specific responses to the reviewer's comments are provided below:

Point 1:  The results should be compared with the state of the art advantages and limitations highlighted. 

Response 1: It is not an easy task to compare the results obtained with the results presented in the literature. The experiments in different articles are not defined in the same way and in the same environment. However, we have extended the comparison in terms of success rate in Section 5.3. For this extension, we have reviewed the papers referenced in Section 1.1. The results shown in these papers have been compared with our results in Table 4.

Point 2: In table 3, it is not clear why the CARLA simulator had lower tavg.  It is also clear that the transfer learning from SUMO to CARLA is not very successful. A deeper discussion is needed why there is this major impact as success falls drastically (compared to table 2).

Response 2: CARLA simulator has a lower tavg because the road in sumo is way longer. We have added an explanation in line 337.

The main reason for this fall of success is the dynamics of the simulator. SUMO does not simulate the dynamics of the vehicle, while CARLA allows a realistic dynamics simulation. We have extended Section 5.4 to have a deeper discussion about this topic.

Point 3: Some other minor corrections:

Line 85 - "increases the difficult of the tasks" -> "increases the difficulty of the tasks"

Line 172 - "scenarios the vehicle will found at intersections" -> "scenarios the vehicle will find at intersections"

Line 221 - "These roads length is 200 m and" -> "The length of these roads is 200 m and"

Line 257 - "has a stop signal in his lane while the adversaries" -> "has a stop signal in its lane while the adversaries"

Line 334 - "being used both the network and the weights." -> "using both the network and the weights.".

Response 3: To improve the quality of our paper, we have used the “Grammarly” tool, which is a writing assistant that reviews spelling, grammar, punctuation and clarity of English text. With this tool, we have checked for spelling errors. In addition, we have sent the paper to english-speaking proofreader to improve the English. There are modifications all over the text that can be tracked using the review tab in Overleaf tool.

Round 2

Reviewer 3 Report

The authors have improved the paper and included some comparisons of the results obtained.

Some minor typos need correction:

Line 83 - "in unseen scenarios An additional" -> "in unseen scenarios. An additional"

Line 370 - "As we can see, our results are on-par" -> "The obtained success rates indicate that our results are on-par"

Line 429 - "Adding dynamics to the vehicles movements add complexity" -> "Adding dynamics to the vehicles movements adds complexity"

Author Response

Dear Reviewer:

Thank you very much for your review of our paper. We sincerely appreciate all valuable comments and suggestions, which helped us to improve the quality of the article.

A minor revision of the paper has been carried out to take all of the corrections into consideration. Specific responses to the reviewer3's comments are provided below:

Point 1:  Some minor typos need correction:

Line 83 - "in unseen scenarios An additional" -> "in unseen scenarios. An additional"

Line 370 - "As we can see, our results are on-par" -> "The obtained success rates indicate that our results are on-par"

Line 429 - "Adding dynamics to the vehicles movements add complexity" -> "Adding dynamics to the vehicles movements adds complexity".

Response 1: Thank you very much for your careful review of our document. We have modified the text according to your suggestions.  
